# A Smart Grid Overvoltage Identification System Associated with Partial Discharge Signal and Dielectric Loss Detection

**DOI:** 10.3390/s23187727

**Published:** 2023-09-07

**Authors:** Guojin Chen, Yucheng Zhu, Zihao Meng, Weixing Fang, Wei Xie, Ming Xu, Wenxin Li

**Affiliations:** 1College of Mechanical Engineering, Hangzhou Dianzi University, Hangzhou 310018, China; 2Anji Intelligent Manufacturing Technology Research Institute Co., Ltd., Hangzhou Dianzi University, Huzhou 313000, China; 3Hangzhou Kelin Electric Co., Ltd., Hangzhou 310011, China

**Keywords:** overvoltage, wide frequency range, dielectric loss, partial discharge

## Abstract

Capacitive equipment refers to its insulation design using the principle of capacitance of electrical equipment, mainly by a variety of different capacitive components in series. Most of the equipment in the substation is capacitive equipment. Once an insulation failure occurs, it will lead to extremely serious consequences. Monitoring grid overvoltage and insulation degradation of capacitive equipment is an effective means to ensure the stable operation of the power system. Therefore, in order to enhance the health management of capacitive equipment, including transformers, bushings, and current transformers, and to mitigate the risk of severe failures, it is imperative to conduct broad-spectrum frequency-domain online monitoring of overvoltages, dielectric losses, and partial discharge. However, the current monitoring work requires the utilization of multiple detection apparatuses. Aiming at the disadvantage that the existing inspection is not well integrated and requires a combination of multiple devices. This paper proposes a smart grid overvoltage identification system that utilizes partial discharge (PD) signals in correlation with dielectric loss detection. The system achieves synchronous detection of dielectric loss and high-frequency partial discharge by synchronously and in real-time acquiring four current signals from the power grid, enhancing the integration level of the hardware system.

## 1. Introduction

With the rapid growth of the smart grid industry and the increasing demand for energy, there has been a significant rise in the voltage level. Large-scale energy storage systems, electric vehicle charging facilities, and numerous innovative industrial equipment have been deployed. Energy storage and management between power grids, along with the aim of minimizing energy loss, necessitate the elevation of voltage levels to meet the growing demand. The voltage levels have increased significantly, and the system capacity has grown rapidly, placing higher demands on the safety and reliability of transmission and substation equipment. Currently, online monitoring is the primary means of operation and maintenance for substation units [1,2]. Monitoring the health status of the power system online mainly involves monitoring the overvoltage of the main network and the insulation strength of capacitive equipment [3,4].

The phenomenon of overvoltage in the electrical grid is characterized by the root mean square of the alternating current voltage at the operational frequency exceeding ten percent of the rated voltage magnitude and persisting for a duration exceeding one minute [5]. The occurrence of overvoltage phenomena is intricately linked to load surges on the grid system and the switching of the grid system’s idle state [6]. There exist two distinct categories of overvoltage phenomena, namely external and internal overvoltage. External overvoltages are commonly associated with lightning and can arise from either direct lightning strikes impacting grid system equipment or lightning strikes that produce a potential in the grid system without directly contacting the equipment [7,8]. Overvoltage caused by direct lightning strikes on grid system apparatus has a duration of microseconds and an amplitude of up to millions of volts, exhibiting pulse characteristics. This type of overvoltage can damage the internal insulation of capacitive devices and lead to ground defects caused by short circuits. On the other hand, overvoltage induced by lightning strikes causes a sharp change in the spatial electromagnetic field, while overvoltage induced on equipment not directly struck by lightning can reach up to ten times the rated voltage level and nearly 10,000 volts, causing severe damage to the internal insulation of capacitive equipment and posing serious safety risks to personnel [9,10]. Internal overvoltages include transient overvoltages [11], operational overvoltages [12,13], and resonant overvoltages [14,15] that are induced by alterations in the internal operation of the power supply system.

The focus of this paper is on the examination of a system primarily intended for current transformers. These transformers can be classified into six distinct operating conditions, namely: Main insulation cable paper crease, capacitor screen moisture, current transformer water moisture, inadequate grounding of the final screen, insufficient contact of the primary-side wiring, and inadequate contact of the secondary-side wiring. The insulation fault operating conditions encompass the main insulation cable paper crease, capacitor screen moisture, and current transformer water moisture. On the other hand, the electrical faults include poor grounding of the final screen, poor contact of the primary-side wiring, and poor contact of the secondary-side wiring.

Various techniques are commonly employed for the identification of faults in current transformers. These methods encompass oil chromatography detection, voiceprint vibration detection, infrared temperature measurement technology, dielectric loss detection, and partial discharge (PD) detection. While oil chromatography detection is known for its accurate detection results, it has certain limitations. Specifically, its detection range is restricted to the assessment of capacitive equipment’s main insulation strength. Additionally, on-site oil testing is necessary, which hinders the acquisition of information regarding the operational conditions of the equipment being tested. The detection of voiceprint vibrations exhibits a lack of specificity, and the precision of the detection outcomes is relatively low. Additionally, the installation environment requirements for detection sensors are stringent, and the method is not really universal. Infrared temperature measurement technology is costly to operate and maintain, and the equipment installation environment requirements are high. However, dielectric loss detection and PD detection can detect insulation faults and electrical faults by collecting electrical signals. Therefore, dielectric loss detection and PD detection are suitable for online monitoring in the outdoor environment of substations. Since the frequency span of these two signals is wide, dielectric loss detection requires collecting the power frequency signal, and PD detection requires collecting the high-frequency 30 MHz current signal. The detection method that integrates these two types of detection belongs to wide-frequency-range detection.

Current transformer leakage currents are superimposed currents of wide-frequency-range signals. Each current signal at a certain frequency can provide information about the characteristics of busbar overvoltage and the primary insulation faults of capacitive equipment [16,17]. In wide-frequency-range online monitoring, the current trend is to Fourier transform the full current signal in order to separate each frequency waveform or signal for data analysis. Due to distinct detection methods and means for each frequency band signal, research in the field of substation online monitoring has tended to concentrate on the fusion of information de-detection across a wide frequency range and multiple dimensions. Some researchers have used the reference device comparison method in conjunction with dielectric loss detection or algorithms to enhance the accuracy of dielectric loss detection defects. However, none has addressed the identification of overvoltage types [18,19]. Others have only collected high-frequency current signals for high-frequency PD detection of power equipment, while ignoring the problem of insulation degradation in capacitive equipment [20]. In recent years, researchers have attempted to combine the detection of dielectric loss and partial discharge (PD) and have made progress, with hardware development and research progressively advancing toward a state of high integration. The integration of multidimensional data is becoming a focal point of industry attention [21,22]. This type of information fusion is increasingly used to diagnose defects in capacitive substation equipment [23,24]. However, the data acquisition of both detection methods is independent, the hardware integration of the acquisition system is low, and with the maturity of big data technology, the digitalization and visualization of the grid system are becoming significant directions of development in the industry [25,26].

Integrated dielectric loss and PD detection can provide insight into the operating status of capacitive equipment. To address the limitations of current multi-dimensional detection technology, such as poor integration and cumbersome device operation and maintenance, this paper proposes a smart grid overvoltage identification system that correlates PD signals with dielectric loss detection for 550 kV level substations. In order to enhance the integration of the acquisition system, this study presents a novel 4-way wide-frequency-range synchronous acquisition device. This device effectively integrates the necessary electrical signals for dielectric loss detection, encompassing reference voltage, end-screen current, reference system interval end-screen current, and a high-frequency partial discharge (PD) signal. This design addresses the shortcomings of current multi-dimensional detection technology, which has poor integration and requires cumbersome device operation and maintenance.

Section 2 of the article focuses on the primary methodological approach, which is wide frequency range detection. Section 3 provides a comprehensive description of the system’s design and composition, while Section 4 presents the results of a field test conducted on the system, along with a summary of the findings. Section 5 summarizes the entire text.

## 2. Wide Frequency Range Signal Detection

This section provides a comprehensive exploration of the two current transformer detection methods employed in wideband domain detection: dielectric loss detection and partial discharge detection.

### 2.1. Dielectric Loss Detection

The process of dielectric loss detection, alternatively referred to as the detection of the main insulation dielectric loss factor under current transformer settings, involves the measurement of the dielectric loss factor of the primary insulation medium. Environmental factors, such as excessive voltage and partial discharge (PD), give rise to energy losses or degradation inside the insulating material, resulting in the dissipation of this energy in the form of heat [27]. Dielectric loss is the energy consumed by the insulation material per unit of time. It will lead to an increase in the temperature of capacitive equipment. If the temperature of the equipment under working conditions exceeds the maximum allowed by the operating temperature, there will be insulation failure. Therefore, it is essential to keep the dielectric loss as small as possible. The dielectric loss is usually detected by inverting the calculation of the busbar voltage combined with the digital spectrogram of the end-screen current to obtain the fundamental wave of voltage and current, calculating the phase difference to determine the dielectric loss factor, and then inverting the calculation of the busbar voltage as shown in Equation (1) [28]:(1)$$\left\{\begin{array}{c} I_{1}=\omega U_{1}C_{1},I_{2}=\omega U_{2}C_{2}\\ U_{1}=I_{1}/\omega C_{1},U_{2}=I_{2}/\omega C_{2}\\ K=I_{1}/I_{2} \end{array}\right.,$$
where ω represents the angular frequency of the current, $C_{1}$ and $C_{2}$ denote the equivalent capacitance formed by the main insulation of capacitive devices. K is the primary- and secondary-side variation ratio, where $U_{1}$:$U_{2}$ can be equated to K through a series of equivalents. Since $U_{1}$ is proportional to $U_{2}$, the integration of $I_{2}$ yields $U_{2}$, and then $U_{1}$ is determined through the variation ratio relationship.

The bus voltage basic waveform is derived by applying the fast Fourier transform to the digital spectrum of the bus voltage, which is acquired by the inverse calculation of dielectric loss detection. The phase difference between the voltage and current waveforms is known as the power factor angle Φ, while the remaining angle is referred to as the dielectric loss angle δ [29]. In an ideal scenario, the phase angle of the current and voltage waveforms should be π/2 when the full current is equivalent to the capacitive current. However, in practice, the full current comprises not only capacitive current but also resistive current, resulting in a phase deviation between the current and voltage waveforms. The tangent of this phase deviation angle, i.e., the ratio of resistive to capacitive current, is known as the resistive to capacitive ratio. This ratio can be used to express the degree of dielectric loss in the main insulation [28,30]. In summary, the phase angle of the current and voltage waveforms should ideally be π/2 when the full current is all capacitive. However, in practice, resistive current is also present, leading to phase deviation and the resistive-to-capacitive ratio. This ratio can be used to express the degree of dielectric loss in the main insulation [31].

The prevailing approach in the industry for assessing the relative dissipation factor entails the computation of the discrepancy between current vectors. This is achieved by utilizing two current transformers linked in parallel to measure the end-screen current, hence obtaining the two dissipation factors. Finally, the difference between the two dissipation factors is obtained to calculate the relative dissipation factor. The relative dielectric loss factor can be used to measure the relative level of main insulation degradation between capacitive equipment at the same interval. By comparing the relative dielectric loss factor between phases in a healthy state, it is possible to determine whether a fault in a phase of capacitive equipment is related to the deterioration of the main insulation. Additionally, monitoring the degree of dielectric loss of the main insulation of each interval capacitive equipment in real time can ensure that the insulation strength of the equipment’s main insulation meets safety standards. Figure 1 shows the schematic diagram for calculating the dielectric loss factor, where the vector diagram is a vector decomposition of the full current Is in phase with U into a resistive current $I_{r}$ and a capacitive current $I_{c}$, and the dielectric loss factor is represented as $\tan\delta =\tan\left(\pi /2-\Phi \right)$.

### 2.2. Partial Discharge Detection

Partial discharges in capacitive equipment can expedite the degradation of the primary insulation, resulting in an increased occurrence of partial discharges. The detection of partial discharge (PD) may be categorized into three main methods: High-frequency PD detection, ultra-high-frequency PD detection, and ultrasonic PD detection. The extraction of small partial discharge (PD) signals poses challenges, but ultra-high-frequency signals within the range of 300–3000 MHz exhibit distinct detecting features and provide a more precise reflection of the PD condition. The ultrasonic signal is susceptible to electromagnetic interference, hence making it possible to utilize ultrasonic PD detection to gather data on the discharge site and provide an accurate representation of the PD scenario. The substation site is characterized by an outdoor setting that is prone to significant electromagnetic interference. Consequently, the UHF signal acquisition device is susceptible to environmental interference, leading to severe distortion in the acquired data. Similarly, the ultrasonic partial discharge (PD) detection data acquisition device is also greatly affected by interference in outdoor environments. As a result, UHF PD detection and ultrasonic PD detection methods are predominantly employed in non-operational settings, such as high-voltage test halls. Consequently, the focus of this research is centered on the detection of high-frequency partial discharges (PD).

## 3. System Design

This section of the chapter deals with the design of the hardware part of the wide-frequency-range data acquisition system. The hardware part of the system is the foundation of the wide-frequency-range monitoring system. It analyzes the working principles of each module in the hardware system as well as the actual requirements, and it introduces the design ideas as well as the specifications of the selected devices.

### 3.1. System Hardware Design Analysis

In a substation, an intelligent electronic device (IED) host is installed to cater to each specific voltage level, while a field acquisition system is implemented for every current transformer present. The field acquisition unit is composed of three main components: The end-screen lead wire, a current sensor with a wide frequency range, and a high-speed acquisition circuit module. The device has the capability to be installed at the grounding position of the end screen of the current transformer’s lead wire. This installation is achieved by utilizing a worn core method, allowing for the installation of a current sensor with a wide frequency range that connects back to the grounding point of the end screen. An IED host is installed within the component cabinet of the substation site. Its purpose is to collect data from each wide-frequency-range monitoring device present on-site. This data is transmitted using the IEC61850 protocol through the substation station server, allowing for the monitoring of data on the wide-frequency-range monitoring software platform. Figure 2 illustrates the principle of data acquisition in this system.

In particular, overvoltage monitoring is achieved by inverting the busbar voltage value from the end-screen current power frequency signal through the busbar voltage inversion technique, thus carrying out real-time monitoring of busbar voltage. The calculation of the relative dissipation factor involves utilizing the reference voltage as the phase reference. This entails determining the tangent of the phase angle from the power frequency signal of the end-screen current and performing the same procedure for the power frequency signal of the reference interval. Finally, the difference between the two-phase angle tangents is used to find the relative dielectric loss factor to achieve dielectric loss detection. Therefore, the hardware system design in this paper requires a wide frequency range bandwidth of 50 Hz to 30 MHz and four acquisition channels, including three industrial-frequency signal input ports and one high-frequency signal input port. The collection of the high-frequency signal from the end-screen current enables the identification of high-frequency partial discharge (PD), and the monitoring of PD occurrences is achieved by the calculation of the high-frequency PD quantity. Figure 3 displays the acquisition signals associated with the detection of overvoltage, dielectric loss, and high-frequency partial discharge.

The aforementioned technique of busbar voltage inversion is employed to acquire the frequency signal of the end-screen current. This is achieved by gathering and preparing the signal and subsequently directing the end-screen current into a small signal amplification circuit. The output of this circuit corresponds to the busbar voltage value multiplied by the current value. The technique of busbar voltage inversion is employed to determine the voltage magnitude of the busbar voltage. This is achieved by multiplying the current value of the final screen current so that the resulting current signal is numerically equivalent to the busbar voltage value. The end-screen current amplification is shown in Equation (2):(2)$$I_{out}\left(t\right)=\frac{T}{C\cdot K_{1}\cdot K_{2}}U_{input}\left(t\right),$$
where the compensation factor C is 100; the current conversion linearity factor $K_{1}$ is 1000; the current-voltage conversion linearity factor $K_{2}$ is 100; the integration time constant T is 0.2; the input current value input to the tiny signal amplifier circuit after conditioning is 10 mA; and the output voltage value after amplification is 500 kV.

### 3.2. System Hardware Design Implementation and Optimization

The monitoring device with a wide frequency range is situated within a container and consists of many components, including a power supply, PT signal sensor, switching power supply, terminal, broadband sensor, broadband signal high-speed acquisition board, signal filter, and terminal interface. The power supply vacancy is classified as level 2 and has a rated current of 3 A. It is designed for controlling and protecting power input at an industrial frequency of 220 V. The PT signal sensor is responsible for converting a 220 V industrial frequency voltage into a 2.5 V output. In this system, the power supply voltage is utilized as the phase reference to enable the comparison of the phase laterally in the end-screen signal. The switching power supply is responsible for converting the alternating current (AC) voltage of 220 V at the industrial frequency into a direct current (DC) voltage of 5 V, specifically for the purpose of powering the monitoring module. The wide frequency range sensor is a non-contact installation that is used to measure the end-screen signal in a once-through-center configuration. The wide frequency range signal high-speed acquisition board performs many activities, such as measuring and analyzing the end-screen current signal, calculating the bus voltage, receiving reference voltage input, conducting high-frequency partial discharge (PD), and processing other data waveforms. The signal filter outputs the sensor’s secondary signal to the signal conditioning module of the high-speed acquisition board after pre-amplification, signal superposition, and filtering. The end-screen interface serves the purpose of facilitating the end-screen lead-down connection. It is equipped with an end-screen open-circuit protector and alarm. Additionally, the end-screen signal is directed back to the grounding point after passing through the wide frequency range sensor. Figure 4 depicts the electrical schematic diagram of the wide-frequency-range signal acquisition hardware system.

To improve the hardware system integration and facilitate the daily maintenance of the system hardware, it is necessary to set the end-screen lead down, wide-frequency-range current sensor, signal acquisition board, and reference voltage interface in the same box. Therefore, it is necessary to solve the signal interference problem and improve the device’s performance against electromagnetic interference. There are two additional factors to consider. Firstly, the hardware device of the wide frequency range data acquisition system is situated at the substation site, which is characterized by a high level of electromagnetic interference. Consequently, it is imperative to augment the anti-interference capabilities of each individual unit of the system hardware. Second, each unit module circuit chip of the acquisition system is a precision device. Power supply voltage input surges and pulses can cause damage to the chip in the acquisition system, and the system needs to prevent occasional input side surges or pulses. Therefore, the system hardware optimization includes the addition of a power surge protector at the input of the power supply voltage, the addition of a 485 communication protection module, and the addition of a metal shield protection cover for the signal acquisition board.

In order to mitigate the occurrence of possible power supply and signal line surges, as well as to counteract abrupt shocks arising from elevated ground potential, our objective is to attain voltage and signal filtering and purification. Furthermore, our objective is to improve the resistance and immunity of the acquisition device in our system against pulse groups. The objective may be accomplished by integrating power surge protectors and 485 communication protection modules at the input of the 220 V power supply and 485 bus, respectively. Please see Figure 5 for the electrical schematic design of the optimized hardware system. The diagram illustrates the representation of the 485 communication protection module as BHQ and the power surge protector as SPD. The newly developed power surge protector has an internal integration of a power supply voltage-sensitive protector and a common mode filter. This design enables the surge protector to survive the power surge effect of severity level 4, as specified in the GB/T17626.5 electromagnetic compatibility test. Figure 6 illustrates the physical diagram of the surge protector. Upon the installation of the 485 communication protection module, the communication bus exhibits enhanced resistance against severe degree 4 pulse group interference, as indicated by the physical diagram of the 485 communication protection module in Figure 7. The installation of the signal acquisition board is carried out within the alloy shielding cover. The shield shell is effectively connected to the chassis for grounding purposes. The grounding equipotential of the high-speed acquisition board provides comprehensive coverage, serving as an electrostatic shield. Additionally, it effectively isolates the signal acquisition board card from any electromagnetic interference originating from the final screen circuit. Figure 8 illustrates the comparative diagram depicting the state of the hardware system before and after undergoing optimization.

### 3.3. System Acquisition Module

The main component of the hardware system is the acquisition system, which is divided into front-end acquisition and back-end acquisition. Front-end acquisition is the current transformer end-screen current access to the hardware system, and the wide frequency range current sensor is the primary component of the front-end acquisition component. Back-end acquisition consists primarily of input signal amplification and conditioning, with the high-speed acquisition board serving as the primary component.

(1)Front-end acquisition

The KLJC28-CT model is a wide-frequency-range current sensor that is specifically designed for detecting current in substation equipment across a wide frequency range. It is capable of accurately measuring current from 50 Hz to 30 MHz, including high-frequency PD current. The sensor employs a precise measurement method that ensures a phase angle difference control of less than one-thousandth of a degree and a milliampere measurement ratio difference of less than 0.05%. With a measurement bandwidth of 50 Hz to 50 MHz, the sensor can accurately measure current amplitudes ranging from 0.1 mA to 2 kA. Additionally, the sensor has an overall protection level of IP65, as depicted in Figure 9, making it suitable for various environmental conditions.

(2)Back-end acquisition

The PCB diagram of the high-speed acquisition motherboard is specifically engineered to enable the detection of signals spanning a wide frequency range, including both low and high frequencies. In order to ensure the integrity of the signal input, the port is provided with protective diodes that facilitate the superposition of signals while minimizing interference. Furthermore, in order to precisely detect feeble electrical currents emitted by zero flux sensors, the printed circuit board (PCB) integrates a meticulously engineered high-speed junction field effect transistor (JFET) operational amplifier, widely recognized for its significant amplification capabilities. The conversion of current signals to voltage signals is achieved by the utilization of the resistance principle. The front end of the system is equipped with a regulated excitation current source, a second-order anti-alias filter, and an automated gain amplifier to accommodate the wide frequency range of the current sensor. For high-frequency PD signals using a high gain bandwidth, low distortion current feedback type high-speed amplifier, and then through the band-pass filtering processing signal, the signal can be significantly amplified, as shown in Figure 10 for the high-speed acquisition board after preparation of the physical diagram. The high-speed ADC sampling control is flexible, allowing for software-customized settings of sampling rate and sampling ratio for different types of signals. Combined with digital filters to modify the signal-to-noise ratio. The signals are separated, processed, and passed to the analog-to-digital signal conversion module. A high-speed differential drive circuit is augmented by a low-voltage noise field effect tube, enhancing its precision and enabling the acquisition of a substantial drive. Moreover, this drive possesses an expansive dynamic range. The ADC digital-to-analog converter chip, shown in Figure 11, is not fully compensated, which enables effective noise reduction and significant bandwidth improvement.

### 3.4. Micro Signal Amplification Module

The minuscule signal amplification circuit designed in this paper needs to have an input voltage range of 5 mV to 5 V and an input current range of 10 mA to 1 A. Additionally, the phase angle difference between the amplified output signal and the input signal must not exceed 0.001 degrees. As the dissipation factor is calculated using the end-screen current signal and the reference voltage signal, this paper also performs a high-frequency bandpass amplification of the tiny signal, wide-frequency-range sensor once through the heart of the way to measure the mA-level leakage current letter, the primary side of the sensor, and the secondary side of the variable ratio of a thousand times the ratio, so the secondary output is a tiny microampere signal. The reference voltage signal conditioning circuit and the end-screen current signal conditioning circuit are depicted in Figure 12 and Figure 13, respectively. In order to monitor the occurrence of overvoltage in the busbar, it is imperative to determine the busbar voltage by analyzing the measured data. This involves measuring the leakage current at the end screen of the current transformer, which is 0.1 mA. This current is then transformed by a current sensor with a wide frequency range to obtain a secondary current of 0.1 μA. Subsequently, the voltage waveform data can be obtained through a passive integration circuit utilizing a detection impedance of 50 ohms. To facilitate further analysis, it is necessary to amplify this signal by a factor of ten thousand, resulting in a 50 mV signal that can be digitized and subjected to signal analysis using an ADC.

This study presents the design of a transimpedance operational amplifier (op amp) circuit that has an integrated bandpass filtering function. The system utilizes a junction field-effect transistor (JEFT) input op amp and an integrated op amp with an ultra-high gain bandwidth of 1.6 GHz. The selection of the amplifier is based on its low noise and high accuracy characteristics, making it suitable for use as a Vocm amplifier model. The integration of a two-stage transimpedance amplifier circuit with a bandpass filter results in a substantial reduction in signal phase hysteresis. This configuration enables the amplification and acquisition of secondary weak signals from high-performance sensors with a high level of precision. The output voltage of the voltage conditioning circuit is shown in Equation (3):(3)$$U_{out}=\frac{U_{0}\cdot R_{u1}\cdot R_{u2}}{2R_{u3}\cdot R_{u4}},$$

The output of the final screen current amplifier circuit is shown in Equation (4):(4)$$I_{out}=\frac{I_{0}\cdot R_{u5}\cdot R_{u6}}{R_{u7}^{2}},$$
where $U_{out}$ is the output voltage, the rated voltage of the acquisition system is 5 V; and $U_{0}$ is the input voltage of 5 mV. If the output voltage meets the rated voltage of the acquisition system, it can be calculated that the resistance value of the sampling resistor $R_{u1}$ is 500 ohms, the resistance value of $R_{u2}$ is 10 k ohms, the resistance value of $R_{u3}$ is 25 ohms, and the resistance value of $R_{u4}$ is 100 ohms. Iout is the output current. The rated voltage of the acquisition system is 1 V; $I_{0}$ is the input current of the final screen is 10 mA; if the output voltage meets the rated current of the system, it is calculated that the resistance values of $R_{u5}$ and $R_{u6}$ are 500 ohms; the resistance value of $R_{u7}$ is 50 ohms.

The output signal parameters of the conditioning circuit are shown in Equations (5) and (6):(5)$$f=\frac{1}{2\pi RC},$$
(6)$$\varphi =\arctan\frac{f_{0}}{f},$$
where f is the output signal output frequency; $f_{0}$ is the input signal frequency; φ is the phase angle difference generated by the conditioning circuit to make the phase angle difference between the input and output signals of the conditioning circuit meet 0.001 degrees and as small as possible, choose the resistance of the conditioning circuit resistance R value of 10 kΩ, C is the capacitance of the conditioning circuit, the capacitance value of 1 nF, the output frequency f can be found at 5 MHz, so the phase angle difference generated in the conditioning circuit The phase angle difference is 0.0006 degrees, which meets the demand. The high-speed wide-frequency signal acquisition board uses a 16-bit high-performance ADC with 4-channel parallel synchronous conversion. This analog input amplifier supports various user-selectable input ranges, flexible and adjustable data acquisition rates from 10 Ksps to 5 Msps, and excellent anti-interference performance of data filtering and sampling, thus realizing high-precision synchronous sampling of multi-dimensional signals. Since the acquisition signal of the local discharge detection is also a small signal, the PD detection module uses a transimpedance op amp with integrated bandpass filtering to amplify the small signal with a high bandpass. 

### 3.5. High-Frequency Partial Discharge Signal Conditioning Module

The partial discharge detection module designed in this paper reduces the output signal of the bus voltage integration circuit to a low-voltage signal for subsequent signal input, and the partial discharge signal conditioning circuit is shown in Figure 14. The high-frequency interference is eliminated with an LC high-pass filter circuit with a cutoff frequency of 50 MHz, and then the signal is shaped and converted to waveform reformat the input signal and output it as a square wave. The next signal for a PLL (Phase Locked Loop) phase-locked loop frequency doubling process, the frequency of 50Hz of the industrial frequency input signal, for sub-divided frequency doubling, the total number of times the multiplication of 3 times and the unit multiplication factor of 100, the total multiplication of 10,000,000 times, so that the phase-locked loop output frequency of 5MHz.The reset signal of the frequency doubling circuit ensures that the input signal and the counter align precisely at the forward zero point simultaneously. The phase value can be determined by replacing the equivalent value of the PD pulse. The purpose of a filtering circuit is to mitigate the impact of high-frequency signal interference that may occur at the input of the frequency signal. In order to mitigate any interference caused by the high-frequency signal in the reference voltage signal, measures are implemented to safeguard against the disturbance of the phase associated with the PD pulses. The filter circuit has a passband gain of 0.02, a quality factor of 0.3356, and a cutoff frequency of 1 MHz, as determined by a 3 dB attenuation.

### 3.6. Highly Integrated FPGA Module

The field-programmable gate array (FPGA) is utilized to perform wide-frequency-range, multi-dimensional signal detection by simultaneously controlling four high-speed analog-to-digital converter (ADC) conversions. These four ADC conversions are executed in parallel and are driven by a phase-locked loop (PLL) for synchronous frequency division and asynchronous storage. The printed circuit board (PCB) is evaluated for both the high-frequency clock wiring delay and the data readback time in the high-frequency analog-to-digital (AD) conversion process. The phase-locked loop (PLL) is utilized to manipulate numerous analog-to-digital (AD) chips, enabling the adjustment of clock phase shifts and other intervals. Consequently, the field-programmable gate array (FPGA) is able to accurately interpret the high-frequency AD data.

In regards to the control of the ADC chip, the system deviates from the prescribed instructions outlined in the AD handbook for programming or the conventional approach to reading AD data. In order to ensure accurate data collection from the external AD chip, it is necessary to generate a clock signal in the FPGA that is synchronized with the AD clock. This synchronization allows for the precise reading of AD data and prevents data loss that may occur when the AD clock is halted due to manual timing requirements. The synchronized clock serves as a means of coordinating the timing of data collection.

The common method for achieving high-speed acquisition in traditional systems is to enhance the acquisition speed through the utilization of clock-doubling techniques. This paper discusses the utilization of a five-way analog-to-digital conversion in parallel to achieve consistent and efficient high-speed acquisition performance at a stable frequency. This approach enables the realization of high-precision, 16-bit high-speed acquisition while simultaneously reducing the system clock frequency requirement. Consequently, the wide-frequency-range signal high-speed acquisition board becomes more suitable for reliable operation in the challenging electromagnetic environment of substations. The FPGA chip is utilized for both signal capture and fast Fourier transform (FFT) signal analysis. The fast Fourier transform (FFT) operating unit implemented on a field-programmable gate array (FPGA) enables the achievement of a highly integrated system for data acquisition and analysis. Simultaneously with its high-speed acquisition capabilities, the chip incorporates a logic unit that employs Fast Fourier Transform (FFT) operation to facilitate the conversion of frequency-domain signals to the time domain. This enables the chip to achieve time-frequency conversion of the same input. Since the system has high-speed ADC data over 100 M, it requires the FFT to run at a speed of 100 M or more. In this study, we propose an optimized FFT algorithm logic that builds upon the conventional FPGA-based FFT scheme. Our design achieves a remarkable increase in speed, reaching up to 150 M, while also demonstrating an efficient 1024-point FFT computation cycle that closely approximates the theoretical value of just 61,960 ns. The field-programmable gate array (FPGA) has internal components such as multipliers and adders, facilitating the construction of the digital signal processing (DSP) module. Additionally, it employs four TRUEDPRAMs to enable extensive pipelining, hence enhancing the speed of Fast Fourier Transform (FFT) computations. The DSP computing function and the complete FFT algorithm module are implemented utilizing a total of 13 hardware multipliers, over 1000 gates, and 800 flip-flops. In contrast to the conventional method of storing and transmitting a substantial volume of data to a specialized DSP chip, the approach presented in this study offers advantages in terms of conserving system resources, enhancing signal processing velocity, minimizing network transmission of extensive data, optimizing communication efficiency, and augmenting both FPGA operation and computational speed. The FPGA device selected for this study is the CycloneV series chip manufactured by Intel. This particular chip offers notable benefits such as enhanced integration, exceptional performance, and reduced power consumption.

### 3.7. Power Modules

The power supply module designed in this paper requires an input voltage of 9 to 13.6 V to be stepped down to obtain the 2.5 V required by the FPGA and the 1.2 V required by the digital-to-analog converter ADC, respectively. The acquisition board is the main component of the battery-powered portable instrument’s power consumption, so it is necessary to reduce the board’s power consumption and improve the power supply’s efficiency. Simultaneously, it is imperative for the design to take into account the requirements of shielding noise reduction and anti-interference for the high-frequency signal-detecting channel. The DC-DC module exhibits a notable level of efficiency during step-down operations, albeit necessitating a minimum voltage difference of 5.5 V between input and output. On the other hand, the low dropout regulator (LDO) demands a smaller minimum voltage difference of 2.5 V, but its step-down efficiency is comparatively lower, leading to a propensity for significant heat generation.

The power supply input for the high-speed acquisition board in this study is designed by combining a high-efficiency DC-DC module with a low-ripple LDO power supply circuit. The main power supply is designed to have high efficiency, high output current, low quiescent current, and a minimal number of peripheral circuit inductors and filter capacitors, based on the characteristics of the power supply module. According to the data presented in Figure 15, the input voltage range for the dual DC-DC module is 9 to 13.6 V. The module provides two output voltages, namely 5 V and 3.3 V. Each circuit of the module has a maximum output current of 3 A. After implementing power supply noise reduction by changing the output of the switching power supply, a low-noise LDO-TPS70402 power control chip is employed for voltage regulation with effective noise filtering. The power regulator chip demonstrates remarkable noise suppression capabilities and possesses characteristics of reduced dropout. Furthermore, the minimal quiescent current consumption of the device is demonstrated in Figure 16a. The diagram illustrates the effective conversion of the 5.2 V output to a precise 2.5 V voltage supply by the LDO device, which is well-suited for applications involving FPGA. Moreover, it can be observed from Figure 16b that a voltage of 1.2 V is achieved by the process of dividing the 3.3 V power supply for the digital-to-analog ADC. The power control system as a whole can achieve the following specifications: 50 μV output noise/60 dB PSRR, linear adjustment rate is 0.02%/V, load adjustment rate is 0.001%/mV, input and output voltage difference is only 100 mV, maximum output current, single circuit is 1 A, and high ripple rejection ratio is 85 dB.

The power supply regulator chip selected for the design of this paper is the TPS704 series from Germany’s Instrument Semiconductor, which is highly integrated and does not require too many peripheral components.

## 4. Testing and Analysis

In this section, the hardware equipment of the wide frequency range signal online monitoring system is installed in the actual substation. The practicality of the on-line monitoring system for wide frequency range signals is verified through real-world cases.

### 4.1. Installation

The capacitive equipment is operated in the junction box under the capacitive equipment body in a power outage state. To begin, it is recommended to establish a proper grounding connection for the end screen by connecting it through the adapter terminal. Next, it is necessary to detach the wiring from the last row of the adapter terminal and proceed to connect the new end-screen grounding wire. Ultimately, the newly installed end-screen grounding wire should be directed from the junction box’s fire-blocking port and securely attached to the extended grounding flat iron lead of the alloy bellows. The iron–nickel alloy tie bundle bellows should be employed to securely fasten them in proximity to the rod grounding flat iron. Additionally, the end-screen lead wire and end-screen grounding wire should be connected to the box. It is imperative to carefully observe the current direction indicated on the end screen and verify that the sensor ground foot is adequately grounded to ensure dependable functioning. The installation method is depicted in Figure 17.

### 4.2. Analysis and Summary of Test Results

The characteristics of different capacitive equipment operating states differ in three aspects: Overvoltage, equipment insulation degradation, and equipment partial discharge. This study analyzed the overvoltage, relative dielectric loss, and high-frequency PD characteristics of several typical capacitive equipment operating states. The substation comprises numerous intervals, with each interval containing three phases, and each phase is installed with a current transformer. When an anomalous condition arises within a specific timeframe of a substation, the model generates the data processing outcomes and records them in the column designated for anomalous states. The output results encompass several key components, namely the timestamp indicating the occurrence of abnormal data, the identification of the specific abnormal interval, the characterization of the abnormal phase, the indication of whether the abnormal condition has been addressed, and the model’s forecasted abnormal state. The operation and maintenance employees of substations can refer to the relevant data charts with the model prediction results in order to assess the operational status of capacitive equipment and develop tailored plans for equipment operation and maintenance.

#### 4.2.1. Poor Contact of Primary-Side Wiring

According to the analysis of the state identification model in Figure 18, it has been observed that there is a poor connection on the primary side of the wiring for the B-phase current transformer in interval No. 50522 of a 500 kV voltage level substation in Zhejiang. The voltage levels across all intervals, as depicted in the diagram, exhibit fluctuation within the range of 500 kV, with an amplitude of less than 50 kV. In general, overvoltage occurs when the amplitude is more than twice the voltage level, but given that the bus voltage fluctuation amplitude is less than 50 kV, it can be concluded that there is no occurrence of overvoltage. Thus, it is evident that the busbar voltage remains stable without any overvoltage phenomena throughout the monitoring period. Specifically, the average fluctuation of the bus voltage in phase B, within the interval No. 50522, ranges between 492 kV and 512 kV, amounting to 502 kV.

After observing the relative dielectric loss as shown in Figure 19, it can be seen that the relative dielectric loss factor of the phase B current transformer of No. 50522 interval is stable in the range of 0.218% to 0.445% of the normal value fluctuation. The relative dielectric loss factor does not exceed 1% of the main insulation strength of the equipment to meet the operating requirements, so the current transformer main insulation is good.

Finally, observing the high-frequency partial discharge as shown in Figure 20, it can be seen that during the monitoring time period, the partial discharge phenomenon occurs continuously in the B-phase current transformer of No. 50522 interval and the instantaneous partial discharge reaches 3200 pc, which is seriously high. Combined with the overvoltage monitoring results and dielectric loss monitoring results, it is judged that the No. 50522 interval B-phase current transformer has a poor primary-side wiring contact fault, which is consistent with the model output results.

#### 4.2.2. Poor Contact of Secondary-Side Wiring

As shown in Figure 21, in the current transformer data visualization interface for each interval of a 500 kV voltage level substation at a certain location, the abnormal reported result of the state recognition model is poor contact of the secondary-side wiring. During the monitoring time from 0:00 on 29 October 2022 to 24:00 on 8 November 2022, the bus voltage of all intervals is stable, and the bus voltage of the phase C current transformer of interval No. 5042 fluctuates from 500 kV to 509 kV with an average value of 508 kV without overvoltage phenomena.

After observing the relative dielectric loss as shown in Figure 22, it can be seen that the average value of the relative dielectric loss factor of the C-phase current transformer in this interval is as high as −270%. The magnitude overflow chart can be read from the data column. It is much higher than the normal state critical value of 1%; the fluctuation amplitude is very large, so the current transformer main insulation degradation degree is very large.

Finally, observing the high-frequency partial discharge as shown in Figure 23, it can be seen that during the monitoring time period, the partial discharge phenomenon occurs occasionally in the C-phase current transformer of No. 5042 interval. The highest instantaneous partial discharge reaches 964 pC, and the average partial discharge is 290 pC, which is a small amount of partial discharge and a low incidence.

The combined partial discharge monitoring results and dielectric loss monitoring results determine that a fault of poor contact of the secondary-side wiring occurred in the phase C current transformer of interval No. 5042, which is consistent with the model output results.

#### 4.2.3. Capacitive Screen Humidity

As shown in Figure 24, the current transformer data visualization interface for each interval of a local 500 kV voltage level substation is shown. Condition identification model abnormality reported result No. 5062 interval A phase current transformer capacitor screen wet. In the monitoring time period, No. 5062 interval A phase sustained overvoltage, overvoltage maximum value of 6114 kV and an average value of 5944 kV. Other intervals are normal busbar voltage fluctuations in the 500 kV range up and down, and the fluctuation amplitude does not exceed 50 kV.

After observing the relative dielectric loss as shown in Figure 25, the average value of the relative dielectric loss factor of the phase A current transformer in this interval is as high as −98%. The amplitude overflow graph can be read from the data column. It is much larger than the normal state threshold value of 1% and fluctuates greatly.

Upon observing the high-frequency partial discharge, depicted in Figure 26, it becomes apparent that the A-phase current transformer within this interval experiences a continuous occurrence of partial discharge throughout the monitoring duration. The maximum instantaneous partial discharge value measured reaches 11,928 pC, with an average value of 11,198 pC. When a sustained partial discharge transpires and the instantaneous value surpasses 1000 pC, it is indicative of an abnormal partial discharge in the equipment, signaling either a failure or an imminent risk of one. The magnitude of the partial discharge is notably considerable, accompanied by a high frequency of occurrence. By correlating the monitoring outcomes obtained from partial discharge and dielectric loss, it is determined that wetting of the capacitive screen occurs in the A-phase current transformer of interval No. 5062, aligning with the model’s output results.

#### 4.2.4. Insulated Cable Paper Wrinkles

As shown in Figure 27, for the current transformer data visualization interface of each interval of a local 500 kV voltage level substation, the abnormal reported result of the state recognition model is the folded insulated cable paper of the current transformer of phase C of interval No. 5033. In the monitoring time period, No. 5033 interval C phase appeared twice with mutation overvoltage, with an overvoltage amplitude maximum value of 1005 kV; other intervals are normal.

After observing the relative dielectric loss as shown in Figure 28, it can be seen that the relative dielectric loss factor of the phase C current transformer of No. 5033 interval changes abruptly with the sudden change of overvoltage. The peak value of the relative dielectric loss factor is 1.18% over the normal state critical value, and when the overvoltage ends, the relative dielectric loss factor returns to normal.

Finally, the observation of high-frequency partial discharge as shown in Figure 29 shows that while overvoltage occurs, the C-phase current transformer of this interval undergoes a sudden-change partial discharge phenomenon. The peak value of instantaneous local discharge reaches 1225 Pc, which exceeds the normal condition threshold value of 1000 Pc. Combining the results of partial discharge monitoring and dielectric loss monitoring, it is judged that the capacitive screen wetting occurs in the phase C current transformer of the No. 5033 interval, which is consistent with the model output results.

#### 4.2.5. Fault Risk State

Figure 30 shows the visualization interface of current transformer data for each interval of a 500 kV voltage level substation at a site. The abnormal results cannot be recognized by the model’s automatic output as the No. 5033 interval current transformer needs to be focused on. During the monitoring time period, all three phases of this interval sustained high-amplitude overvoltage.

Take the interval A phase as an example, observe the relative dielectric loss as shown in Figure 31. No. 5033 interval A phase current transformer’s relative dielectric loss factor is 0.079–0.32% between the normal value range fluctuations. When the relative dielectric loss factor does not exceed 1%, the strength of the main insulation of the equipment meets the working requirements. The main insulation of the current transformer is good.

Following the inspection of the high-frequency partial discharge, depicted in Figure 32, it is evident that the occurrence of partial discharge in the phase A current transformer is within interval No. 5033. As previously mentioned, when equipment endures sustained partial discharge and the local discharge surpasses 1000 pC, it is considered an abnormality. However, the maximum peak value of the instant partial discharge in the phase A current transformer of interval No. 5033 is merely 584 pC, which is below the critical threshold. Despite the low incidence of partial discharge, the extreme value of instant partial discharge has increased from 434 pC to 584 pC over a specific timeframe, indicating an aggravation in the partial discharge phenomenon during this period. Hence, the local current mutual inductance of phase A within interval No. 5033 has not reached an abnormal state. By integrating the observations from local discharge monitoring and dielectric loss monitoring, it can be concluded that a three-phase high-amplitude overvoltage exists in interval No. 5033, although it has not resulted in any damage to the three-phase current transformer. Hence, it warrants attention.

#### 4.2.6. Normal State of Operation

Displayed in Figure 33 is the data visualization interface of the current transformer for each 500 kV voltage level substation interval. Throughout the monitoring period, the three-phase bus voltage of each interval experienced minor fluctuations within a narrow range slightly above or below 500 kV. These fluctuations remain within 50 kV, thus affirming the stability of the bus voltage at the substation and the absence of any overvoltage occurrences.

Observation of the relative dielectric loss as shown in Figure 34 shows that all intervals of the substation three-phase current transformer relative dielectric loss factor are normal and stable with a small range of fluctuations. The maximum value of 0.652% is less than the normal state threshold of 1%, so all the intervals of the substation are in the normal state.

Upon careful examination of the frequent occurrence of partial discharge, as illustrated in Figure 35, it becomes apparent that the phenomenon of partial discharge in all intervals of the three-phase current transformer in the substation takes place. However, it is noteworthy that the peak value of the instantaneous partial discharge does not exceed 1000 pC, and there is a dearth of continuous partial discharge with a low discharge rate. If the partial discharge frequency remains low and the overall discharge remains at a low level, it can, in turn, be inferred that the operation of the current transformer is functioning optimally.

### 4.3. Test Analysis Summary

Conclusion of the Test Results in Conjunction with the Output of the Current Transformer State Identification Model:(1)Normal condition:

The following conditions should be met:

The absence of overvoltage is seen, and the bus voltage exhibits fluctuations around 500 kV, with the amplitude of these fluctuations not exceeding 50 kV. It is imperative that the dielectric loss factor remains below the threshold of 0.132% on average. The maximum allowable value for instantaneous partial discharge is 1309 pc, but the average value of the local party should be kept below 711 pc.

(2)Insulated cable paper fold monitoring data:

The following characteristics should be observed:

An abrupt increase in voltage occurs, surpassing an amplitude of 988 kilovolts. Sudden variations in the dielectric loss factor are seen, with fluctuation amplitudes exceeding 57 times the average value of the dielectric loss factor. Sudden variations in the high-frequency partial discharge (PD) are seen, resulting in the instantaneous local square volume exceeding 933 picocoulombs.

(3)Capacitor screen humidity monitoring data:

The following characteristics should be observed:

Continuous overvoltage occurs, and the overvoltage amplitude exceeds 2031 kV. Large fluctuations in the dielectric loss factor are observed, the average value of the dielectric loss factor exceeds 0.983%, and the fluctuation amplitude is higher than 5.41%. The instantaneous local discharge is higher than 1973 pc.

(4)Current transformer inlet moisture monitoring data:

The following characteristics should be observed:

There is a continuous overvoltage, and the overvoltage amplitude is higher than 721 kV but not more than 1058 kV. The average value of the dielectric loss factor is higher than 0.473%, and the fluctuation amplitude is greater than 4.77%. The instantaneous local discharge is not higher than 1157 pc.

(5)End-screen grounding defect monitoring data:

The following characteristics should be observed:

There is a continuous overvoltage, and the overvoltage amplitude is higher than 1295 kV. The average value of the dielectric loss factor exceeds 0.937% but does not exceed 5.17%. The instantaneous local discharge exceeds 1587 pc.

(6)Primary-side wiring poor contact monitoring data:

No overvoltage occurs, and the bus voltage fluctuates around 500 kV with the fluctuation amplitude not exceeding 50 kV. The average value of the dielectric loss factor should not exceed 0.414%. The instantaneous PD exceeds 1667 pc, and the average value of high-frequency PD exceeds 1184 pc.

(7)Secondary-side wiring poor contact monitoring data:

No overvoltage occurs, and the bus voltage fluctuates around 500 kV with the fluctuation amplitude not exceeding 50 kV. The average value of the dielectric loss factor exceeds 0.687%, and the amplitude of the dielectric loss factor exceeds 5.98%. The instantaneous local discharge does not exceed 1011 pc.

## 5. Conclusions

The operational condition of capacitive equipment, such as current transformers, can be deduced by analyzing variations in its output signal, specifically the current signal observed at the end screen. In order to comprehensively evaluate the operational status of the equipment, it is necessary to combine low-frequency signals (dielectric loss detection) and high-frequency signals (high-frequency partial discharge detection) with overvoltage scenarios. This is because the end-screen current signal exists in a superposition state across a wide frequency range. This methodology offers a more extensive comprehension of the operational status of capacitive equipment.

In this study, we designed a high-speed data acquisition card that combines information on industrial frequency faults in the full current of the end screen obtained by dielectric loss detection with information on high-frequency faults obtained by high-frequency PD detection. The card allows for the simultaneous real-time acquisition of four current signals from the power grid, including a high-frequency PD signal, the end-screen current of the capacitive equipment under test, the reference voltage and end-screen current of the reference equipment, and three channels for dielectric loss detection. The anti-interference capability of the acquisition device was enhanced by incorporating a power surge protector, a 485 communication protection module, and a metal shield for the signal acquisition board. This improvement renders the device more suited for hardware acquisition in environments with high levels of electromagnetic interference. Through actual fault cases, we analyzed the test results and summarized the environmental requirements for using the wide frequency range signal online monitoring system designed in this paper.

## Figures and Tables

**Figure 1 sensors-23-07727-f001:**
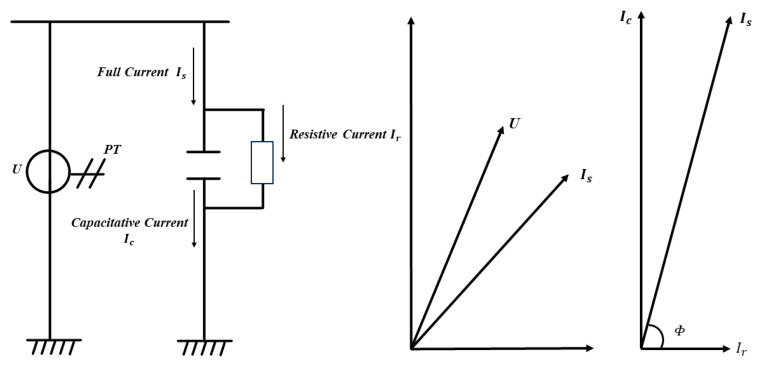
Dielectric loss calculation principle.

**Figure 2 sensors-23-07727-f002:**
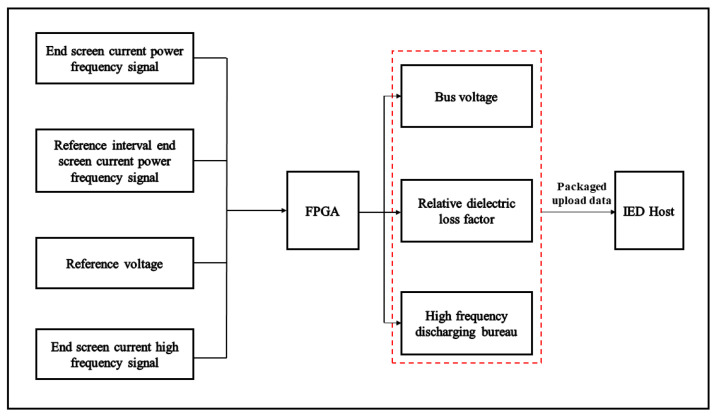
Data acquisition principles.

**Figure 3 sensors-23-07727-f003:**
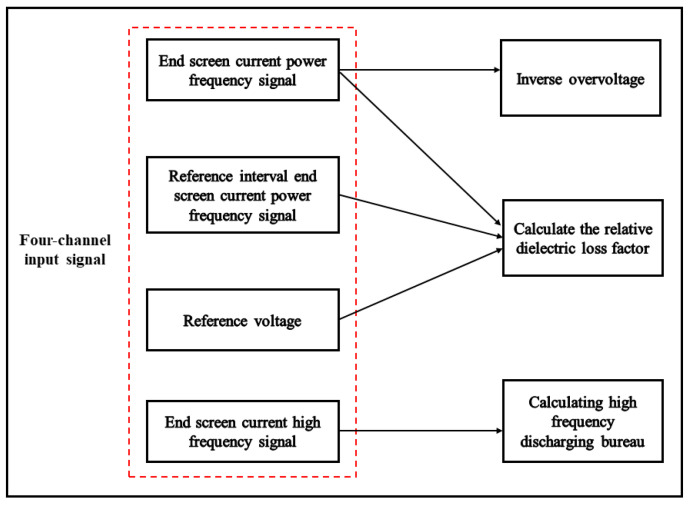
Correspondence between detection type and acquisition signal.

**Figure 4 sensors-23-07727-f004:**
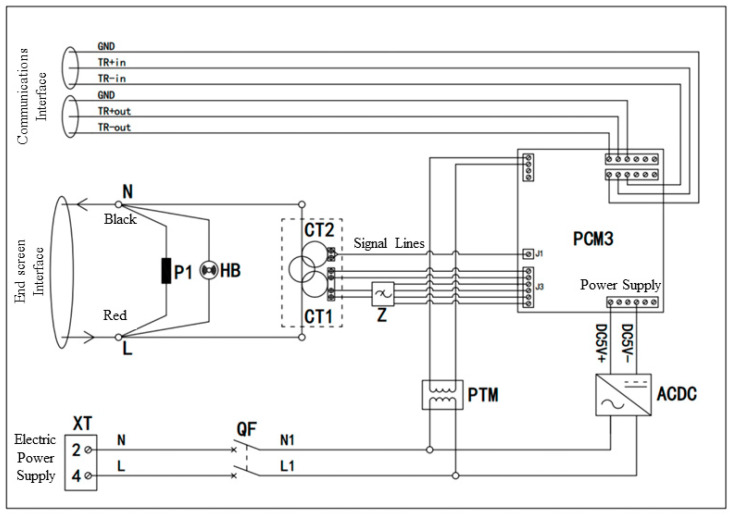
Hardware system electrical schematic. (P1: anti-open circuit protector; HB: open circuit protector; PCM3: wide-frequency-range signal monitoring module; CT1: wide-frequency-range sensor; CT2: partial discharge sensor; PTM PT: signal sensor; Z signal filter; ACDC: power switches).

**Figure 5 sensors-23-07727-f005:**
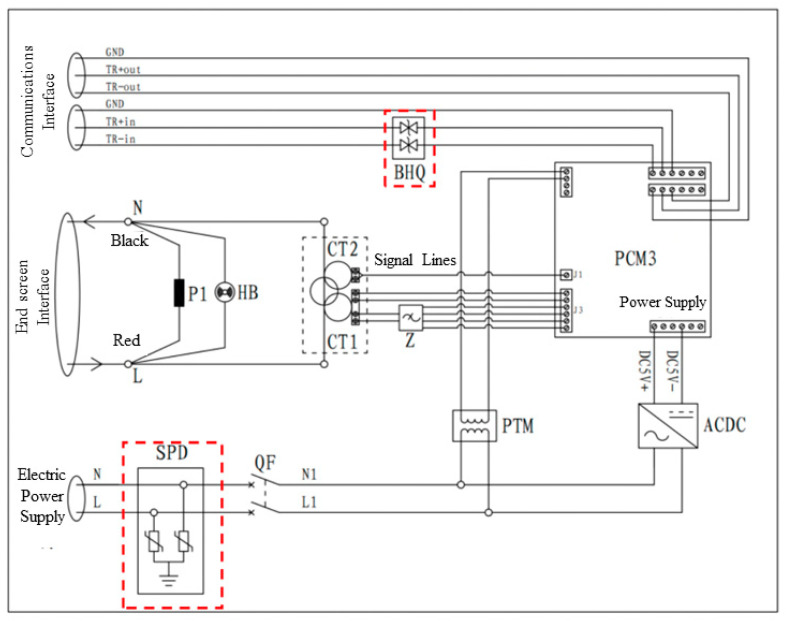
Electrical schematic diagram after hardware system design optimization.

**Figure 6 sensors-23-07727-f006:**
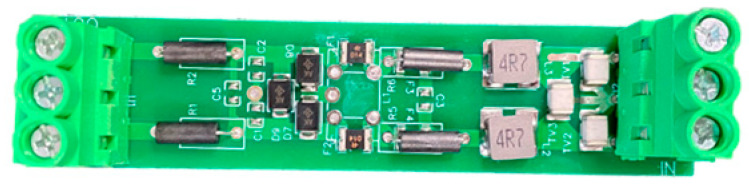
SPD power surge protector design after preparation of physical drawings.

**Figure 7 sensors-23-07727-f007:**
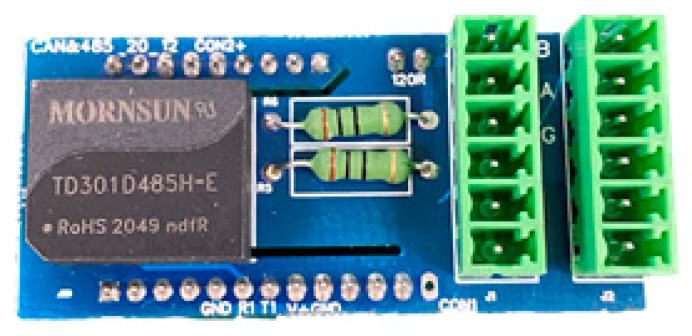
Physical drawing of BHQ485 communication protection module after design and preparation.

**Figure 8 sensors-23-07727-f008:**
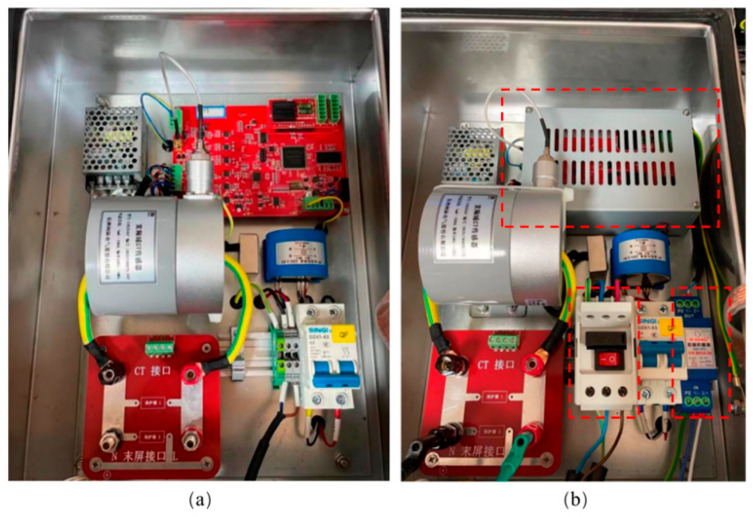
Electrical schematic diagram after hardware system design optimization. Comparison of hardware system design before and after optimization. (**a**) Before optimization; (**b**) after optimization.

**Figure 9 sensors-23-07727-f009:**
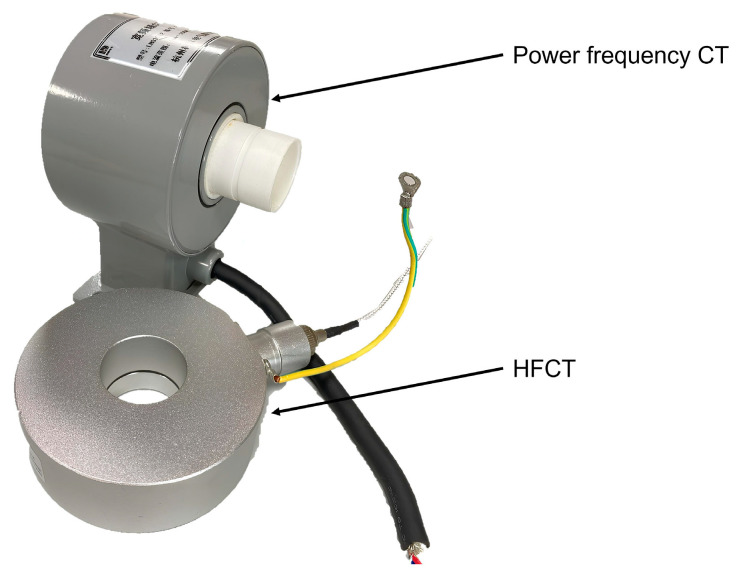
Wide-range current sensor physical diagram.

**Figure 10 sensors-23-07727-f010:**
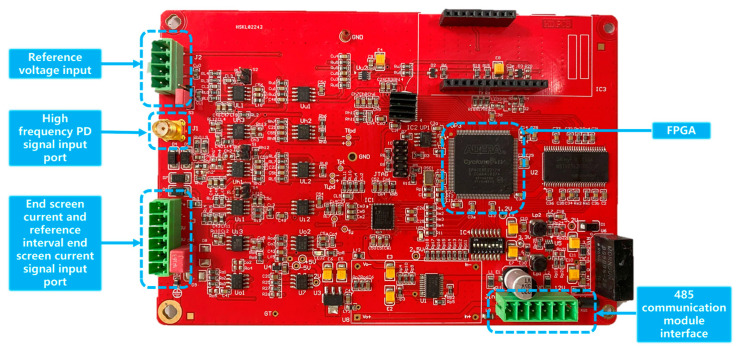
Physical drawing of the high-speed acquisition board after design and preparation.

**Figure 11 sensors-23-07727-f011:**
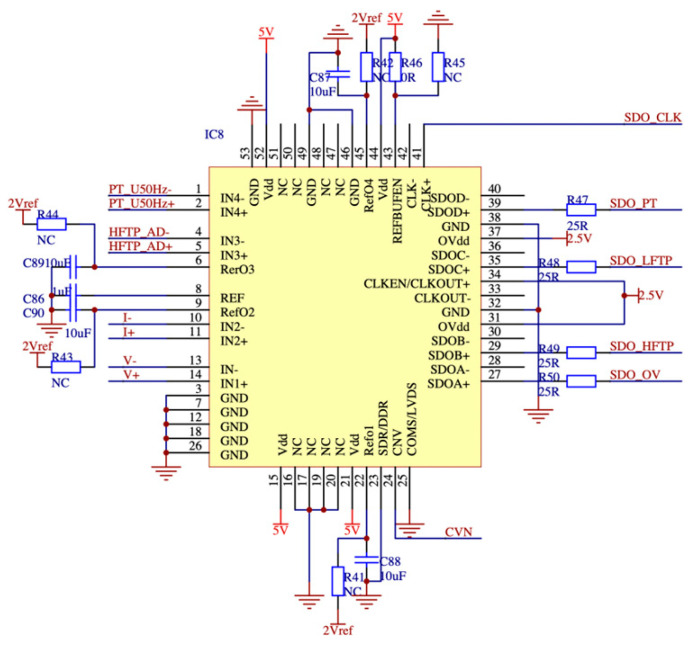
ADC digital-to-analog converter chip design diagram.

**Figure 12 sensors-23-07727-f012:**
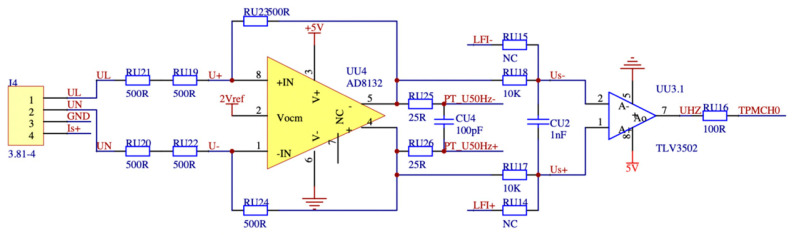
Reference voltage signal conditioning circuit design diagram.

**Figure 13 sensors-23-07727-f013:**
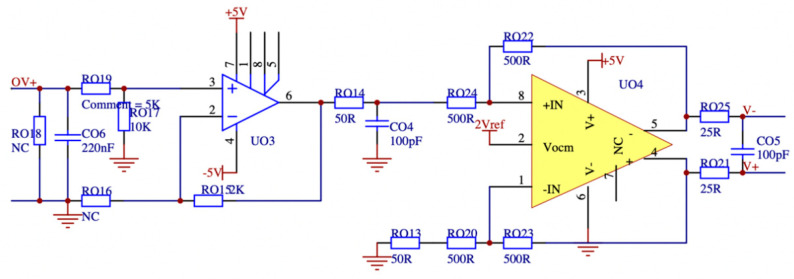
Final screen current signal conditioning circuit design diagram.

**Figure 14 sensors-23-07727-f014:**
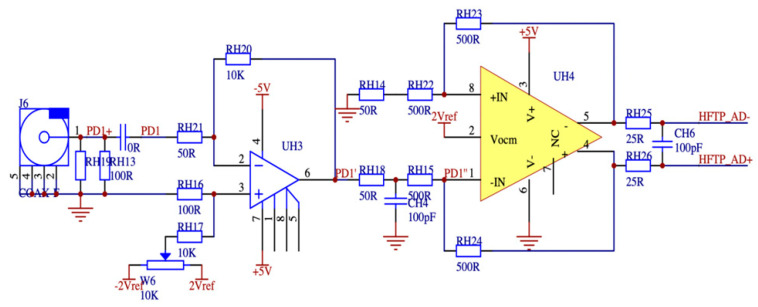
Partial discharge signal conditioning circuit design diagram.

**Figure 15 sensors-23-07727-f015:**
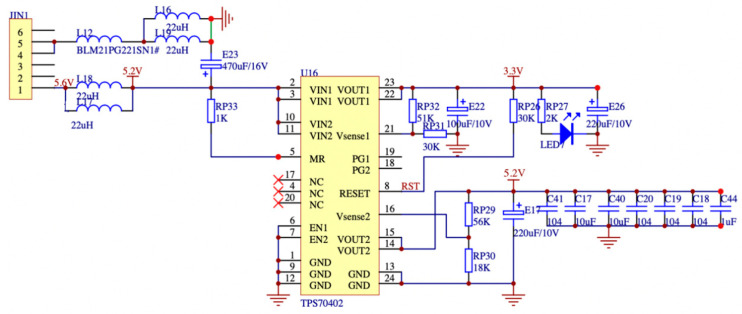
DC-DC power supply circuit design.

**Figure 16 sensors-23-07727-f016:**
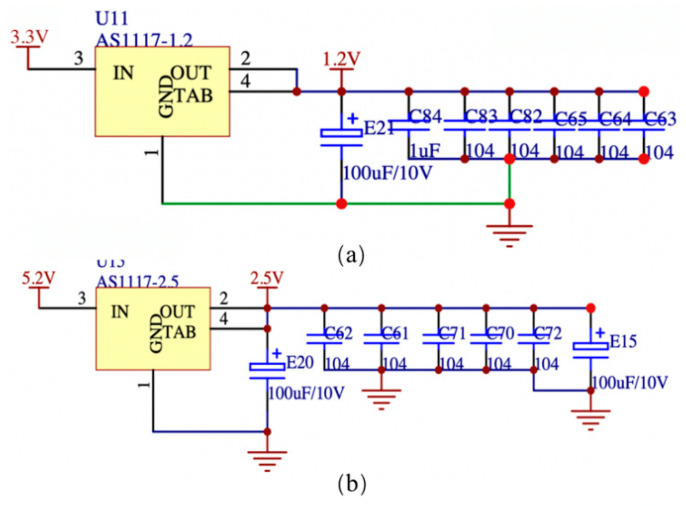
LDO power supply circuit design diagram (**a**) 3.3 V to 1.2 V step-down circuit; (**b**) 5.2 V to 2.5 V step-down circuit.

**Figure 17 sensors-23-07727-f017:**
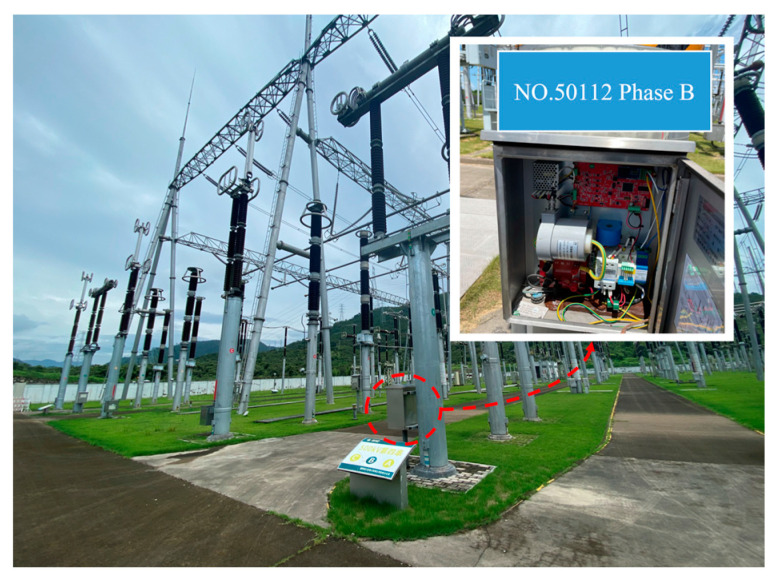
Wide-range current sensor physical diagram.

**Figure 18 sensors-23-07727-f018:**
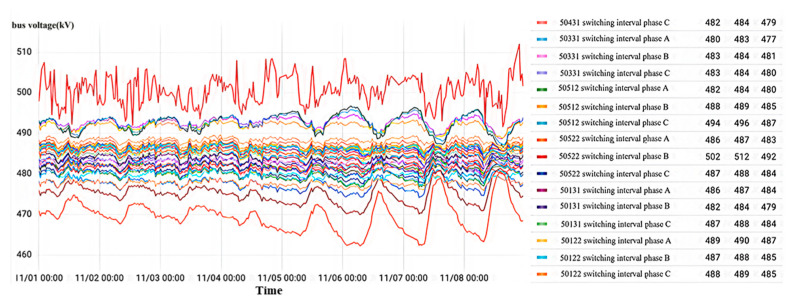
Overvoltage monitoring in a substation A in Zhejiang.

**Figure 19 sensors-23-07727-f019:**
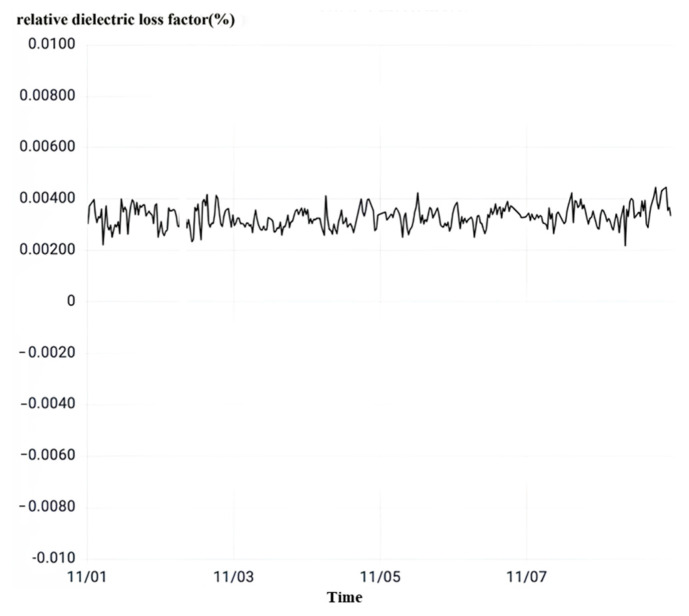
No. 50522 interval rheological relative dielectric loss.

**Figure 20 sensors-23-07727-f020:**
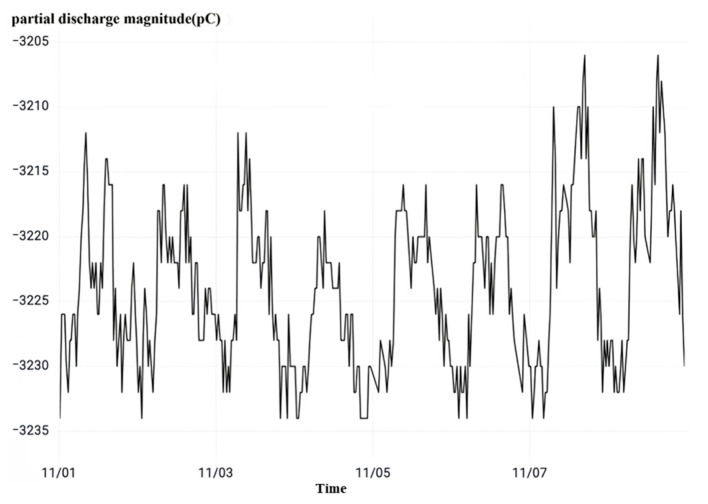
No. 50522 interval high-frequency partial discharge.

**Figure 21 sensors-23-07727-f021:**
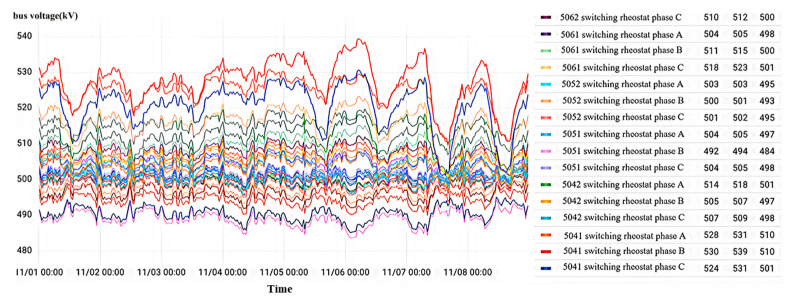
Overvoltage monitoring in a substation B in Zhejiang.

**Figure 22 sensors-23-07727-f022:**
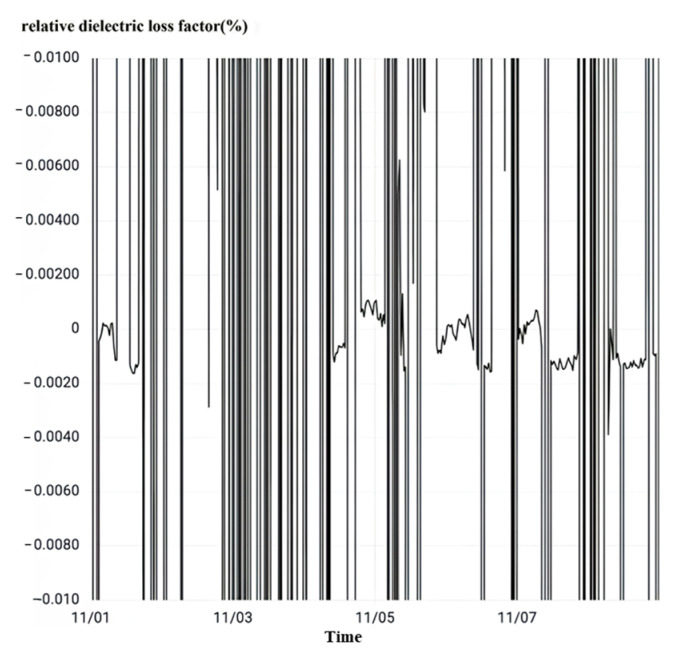
No. 5042 interval rheological relative dielectric loss.

**Figure 23 sensors-23-07727-f023:**
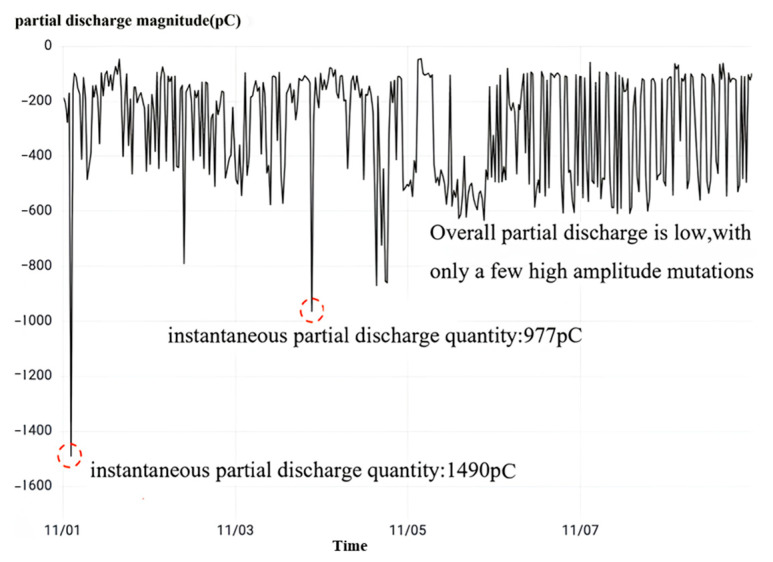
No. 5042 interval rheostat high-frequency partial discharge.

**Figure 24 sensors-23-07727-f024:**
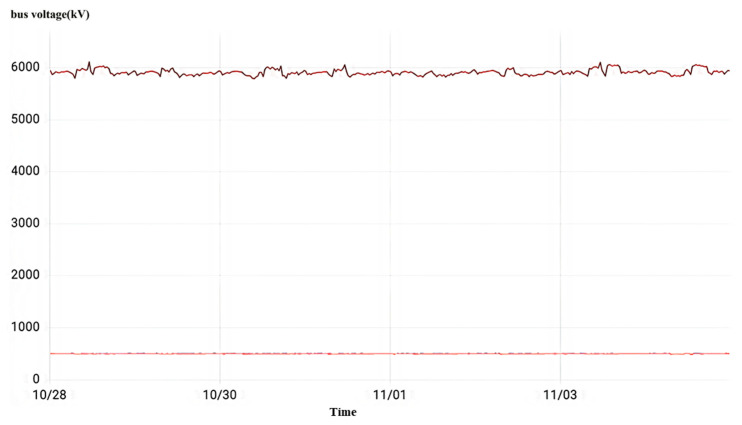
Overvoltage monitoring in a substation C in Zhejiang.

**Figure 25 sensors-23-07727-f025:**
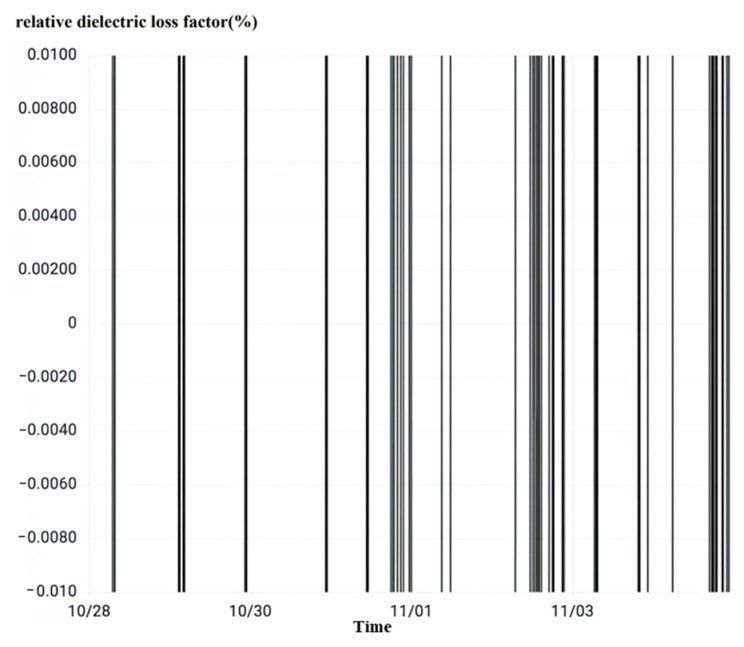
No. 5062 interval A phase rheological relative dielectric loss.

**Figure 26 sensors-23-07727-f026:**
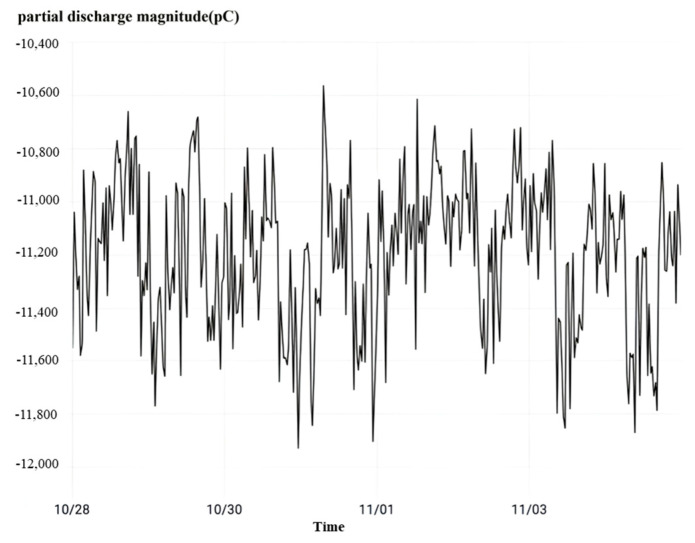
No. 5062 spaced A-phase rheostat high-frequency partial discharge.

**Figure 27 sensors-23-07727-f027:**
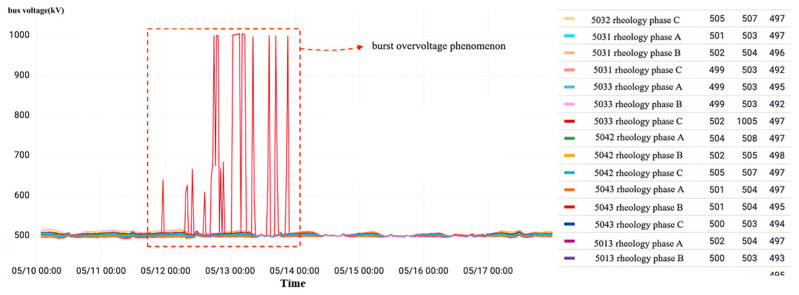
Overvoltage monitoring in a substation D in Zhejiang.

**Figure 28 sensors-23-07727-f028:**
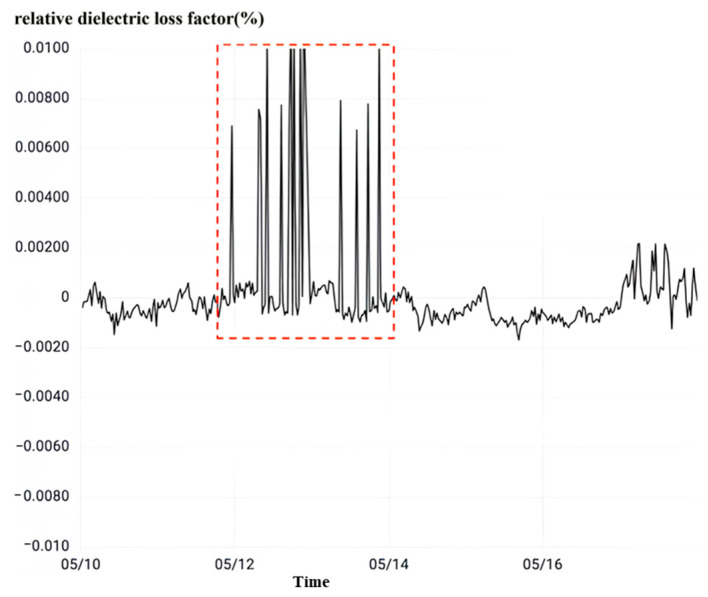
Overvoltage monitoring in a substation E in Zhejiang.

**Figure 29 sensors-23-07727-f029:**
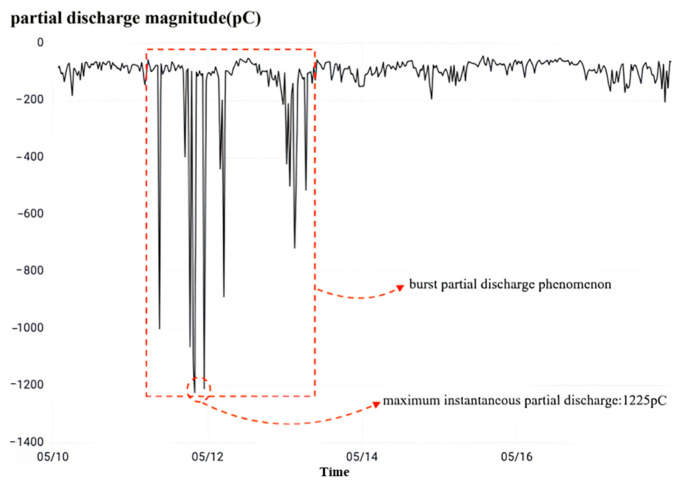
No. 5033 spaced C-phase rheostat high-frequency partial discharge.

**Figure 30 sensors-23-07727-f030:**
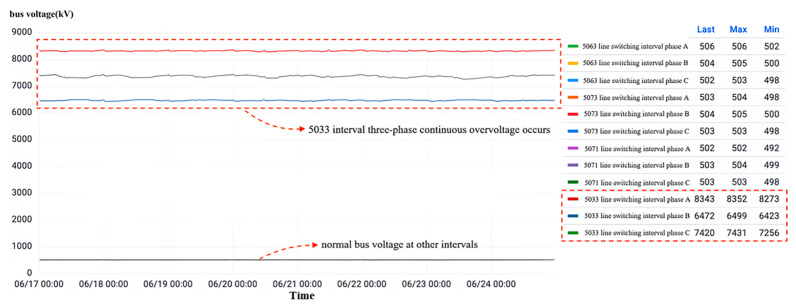
Overvoltage monitoring in a substation F in Zhejiang.

**Figure 31 sensors-23-07727-f031:**
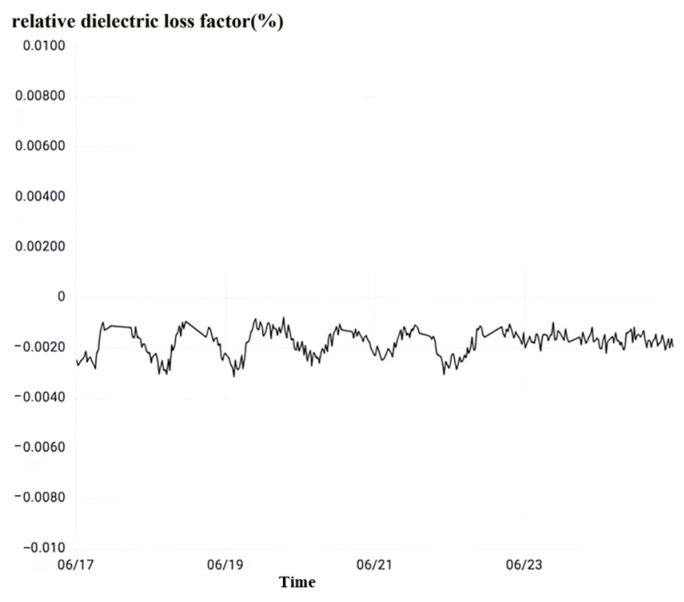
No. 5033 interval A phase rheological relative dielectric loss.

**Figure 32 sensors-23-07727-f032:**
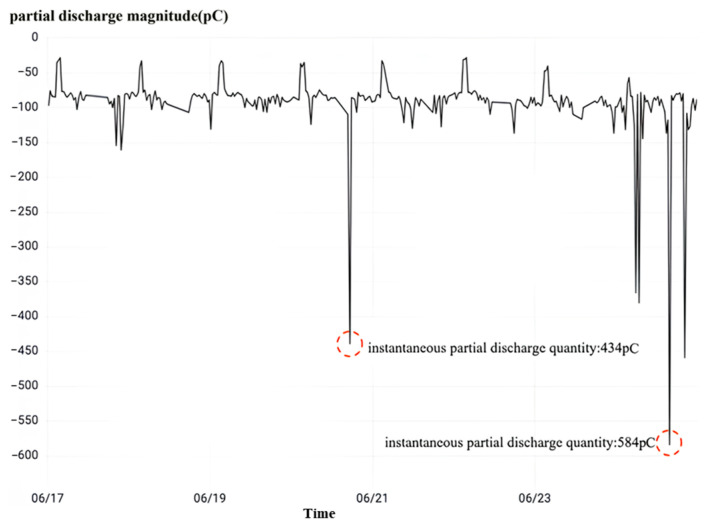
No. 5033 interval A phase rheological high-frequency partial discharge.

**Figure 33 sensors-23-07727-f033:**
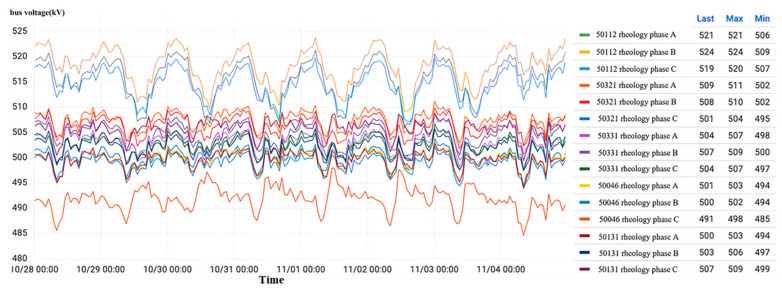
Overvoltage monitoring in a substation G in Zhejiang.

**Figure 34 sensors-23-07727-f034:**
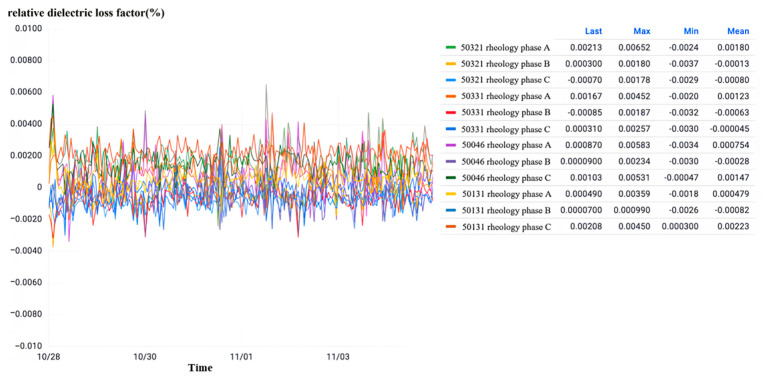
Relative dielectric loss monitoring in a substation G in Zhejiang.

**Figure 35 sensors-23-07727-f035:**
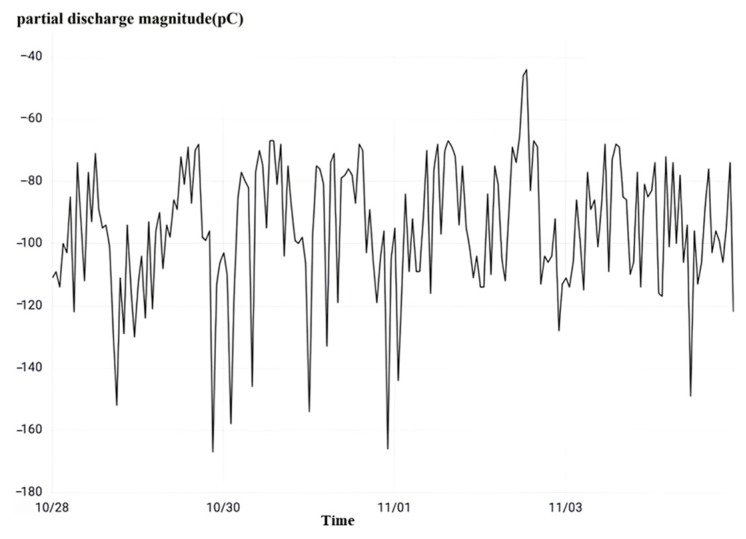
High-frequency partial discharge monitoring at a substation.

## Data Availability

Not applicable.

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
