# Peer review of "A Smart Grid Overvoltage Identification System Associated with Partial Discharge Signal and Dielectric Loss Detection"

_sensors, 2023, doi:10.3390/s23187727_

Round 1
Reviewer 1 Report
This study analyzes the characteristics of different capacitive equipment operating conditions, including overvoltage, equipment insulation degradation, and equipment partial discharge. The results show that the proposed method can accurately identify abnormal states of capacitive devices. For normal conditions, the bus voltage fluctuates around 500kV, there is no overvoltage, and the insulation loss factor and partial discharge are within acceptable ranges. For insulated cable paper fold monitoring, sudden overvoltage, insulation loss factor fluctuations and high-frequency partial discharges were observed. Capacitor screen humidity monitoring revealed continuous overvoltage, large fluctuations in insulation loss factor, and high transient partial discharges. Current transformer inlet humidity monitoring revealed continuous overvoltages, insulation loss factor fluctuations, and low transient partial discharges. Monitoring of ground faults at the end screen revealed continuous overvoltages, fluctuations in insulation loss factor and low transient partial discharges. These results help determine the operating status of capacitive equipment and create maintenance schedules. In this research, a high-speed data acquisition card is designed, which combines the full current information of the end screen of the capacitive equipment obtained from the dielectric loss detection with the high-frequency fault information obtained from the high-frequency partial discharge detection. The card can collect four current signals from the power grid in real time at the same time, including high-frequency partial discharge signals, the terminal current of the capacitive device under test, the reference voltage, the terminal current of the reference device and three channels for medium detection. Through actual fault cases, the test results are analyzed, and the environmental requirements of the online monitoring system using wide frequency domain signals are summarized.It is recommended to accept manuscripts directly
good
Author Response
Thank you very much for your comments
Reviewer 2 Report
1. in the abstract, two statements are not clear: "capacitive devices such as transformers" and "dielectric detection". Moreover, what is it meant by "four channels" without any prior declaration? After a proper clarification the entire abstract is to be rephrased with added concrete outcomes
2. in the introduction, which "voltage levels have increased significantly"? and what does it have to do with the rapid development of the stated industries?
3. opaque sentences/words/phrases with obscure meanings are often used: to be avoided
4. to add a small guiding paragraph for the rest of the paper
5. cannot the paragraph "common methods...signal detection" be fitted into the survey which is already presented in the introduction?
6. make sure to include all the variables used in each equation while indicating the purpose of each equation insertion
7. figure 1: is entitled as dielectric loss calculation procedure, but it actually denotes a simple KCL concerning one node with an erroneous vectorial representation: to check
8. do not leave blank spaces between paragraphs and sub-paragraphs
9. some acronyms are falsely included in figure 2 and/or in the text
10. figure 3 is to be redrawn
11. make sure to include a standard (common) variables representation for all equations
12. figure 4 needs more clarification (e.g., what is HB, P1, etc.)
13. the succession of figs.5-6-7 is a bit confusion for the readers: to better clearly state a definite relation and/or descriptive paragraph before/after each (despite that existing it is an in-text citation for each, but it is never clear for example how the newly integrated circuitry of Figure 6 for example can withstand the EM test...)
14. figure 9 is to be described and annotated. this is also the case for all similar figures
15. figure 10 contains chinese characters and is not comprehensible
16. to enhance the readability of figure 11
17. are figs.5-6-7 and figure 15 related? if yes, how?
18. figure 16 can be deleted (no significant purpose)
19. the testing part (the most important part) is to be more expanded; to add a flowchart methodology behind the hardware implementation; to add discrete outcomes with graphics; to add more graphs and outputs data
The English is better to be improved. It is not bad, but not perfect also.
Reviewer 3 Report
The paper presents a high-speed data acquisition system for monitoring overvoltages, dielectric losses, and partial discharges based on the analysis of the frequency spectrum of the signal at the monitored equipment. The system can prevent serious malfunctions by identifying high frequencies that suggest the presence of partial discharges.
The manuscript is well-structured, and significant works are cited in the context of the current state of research in the field.
The figures are clear, intuitive, and easy to interpret. However, I suggest the following issues to be reviewed:
· · Introducing a diagram of the acquisition system connection to the monitored equipment
· · In Fig 2 the final block is IED host, not IDE;
· · Specify the meaning of T in rel 2;
· · Pay attention to doubling some groups of words or even paragraphs (R.315-316: and the sampling rate, R 370-376 is repeated in R 376-384);
· · R.363-364 "The rated voltage is 1A"? Maybe Volts;
· · R.375 replaces nanofa with nF or nano Farads;
· · Very long, difficult-to-understand sentences and other wording should be revised. (R.257-263, R.305-308, R.318-320, R,338-340, R.476-478).
The system presented in the paper allows monitoring of the capacitive equipment's operation by analyzing the change in the shape of the output signal, changes due to internal phenomena and not external electromagnetic disturbances. Following the tests, models were identified regarding the condition of the monitored equipment.
Round 2
Reviewer 2 Report
1. in the abstract, the term "capacitive equipment, including transformers" is still confusing: generally, transformers posses the characteristics of inductive devices. What is it meant by "capacitive equipment"? Additionally, the abstract did not reveal previous meaning conflicts, such as "four channels"; you begin by the importance of "overvoltage monitoring" (as background) then pose the obstacle of "current monitoring methods" (as a problem statement)...
2. The implementation of renewable energy resources and others, empower the total electrical power generated, not the "voltage quantities"
3. Use "section" not "chapter" when referring to the manuscript's different parts. You have forgotten Section 5
4. The majority of the sentences are still opaque, as is the case during my first review: more effort is to be truly taken into consideration, and to revise the entire manuscript, word-wise
5. Again, the paragraph "Common methods...signal detection" is perceived to be better fit into the Introduction, which already contain a small literature review. How do you see that it is better placed where it is?
6. Figures 1, 2, and 3 are completely blank/non-inserted
7. Define ALL equations' variables: e.g., what is w, C1, C2 in Eq.(1)?
8. Blanks still exist between main/sub paragraphs without any proper explanation/introduction
9. What can a reader understand from checking Figure 10 at a first glance?
10. Explain how necessary is the existence of Figure 16? With the way that you have inserted it, what information does it give to the reader in terms of the overall workflow of your suggested methodology?
English MUST be polished. Avoid the usage of paragraph-like sentences.
Round 3
Reviewer 2 Report
Thank you for finally addressing each of the comments.
Before processing to publication, I kindly have a final small request:
to polish the images where needed, especially ones containing X-Y axis/units. For example, Figure 18 to the last one, has each very narrowed numbers/units. It is required to enlarge these images, sharpening their DPI index and add numbers/units of big fonts.
Better than before.
Author Response
Dear Reviewer:
Thank you again for your professional and rigorous evaluation of this article. The picture has been reprocessed and uploaded according to your requirements.
Best regards
Guojin Chen,Yucheng Zhu, Zihao Meng, Weixing Fang,Wei Xie,Ming Xu,Wenxin Li
09 03,2023